# Effects of Maternal Nutrient Restriction and Melatonin Supplementation on Cardiomyocyte Cell Development Parameters Using Machine Learning Techniques

**DOI:** 10.3390/ani12141818

**Published:** 2022-07-16

**Authors:** Mitra Mazinani, Zully E. Contreras-Correa, Vahid Behzadan, Shreya Gopal, Caleb O. Lemley

**Affiliations:** 1Department of Animal and Dairy Sciences, University of Idaho, Moscow, ID 83844, USA; mitram@uidaho.edu; 2Department of Animal and Dairy Sciences, Mississippi State University, Starkville, MS 39762, USA; zec30@msstate.edu; 3Department of Electrical and Computer Engineering and Computer Science, Tagliatela College of Engineering, University of New Haven, West Haven, CT 06516, USA; vbehzadan@newhaven.edu (V.B.); ssund2@unh.newhaven.edu (S.G.)

**Keywords:** melatonin, cardiomyocyte, fetal development, maternal undernutrition, binucleated

## Abstract

**Simple Summary:**

The major objectives of this study were to examine the effects of maternal feed restriction and melatonin supplementation on fetal cardiomyocyte cell development parameters and predict binucleation and hypertrophy using machine learning techniques using pregnant beef heifers. Interestingly, major findings include evidence that compromised pregnancy in cattle leads to a reduction in the number of cardiomyocytes while melatonin treatment can mitigate some of these disturbances.

**Abstract:**

The objective of the current study was to examine the effects of maternal feed restriction and melatonin supplementation on fetal cardiomyocyte cell development parameters and predict binucleation and hypertrophy using machine learning techniques using pregnant beef heifers. Brangus heifers (n = 29) were assigned to one of four treatment groups in a 2 × 2 factorial design at day 160 of gestation: (1) 100% of nutrient requirements (adequately fed; ADQ) with no dietary melatonin (CON); (2) 100% of nutrient requirements (ADQ) with 20 mg/d of dietary melatonin (MEL); (3) 60% of nutrient requirements (nutrient-restricted; RES) with no dietary melatonin (CON); (4) 60% of nutrient requirements (RES) with 20 mg/d of dietary melatonin (MEL). On day 240 of gestation, fetuses were removed, and fetal heart weight and thickness were determined. The large blood vessel perimeter was increased in fetuses from RES compared with ADQ (*p* = 0.05). The total number of capillaries per tissue area exhibited a nutrition by treatment interaction (*p* = 0.01) where RES-MEL increased capillary number compared (*p* = 0.03) with ADQ-MEL. The binucleated cell number per tissue area showed a nutrition by treatment interaction (*p* = 0.010), where it was decreased in RES-CON vs. ADQ-CON fetuses. Hypertrophy was estimated by dividing ventricle thickness by heart weight. Based on machine learning results, for the binucleation and hypertrophy target variables, the Bagging model with 5 Decision Tree estimators and 3 Decision Tree estimators produced the best results without overfitting. In the prediction of binucleation, left heart ventricular thickness feature had the highest Gin importance weight followed by fetal body weight. In the case of hypertrophy, heart weight was the most important feature. This study provides evidence that restricted maternal nutrition leads to a reduction in the number of cardiomyocytes while melatonin treatment can mitigate some of these disturbances.

## 1. Introduction

Maternal nutrition during pregnancy and fetal growth is strongly associated with determining pregnancy success, health, and productivity of offspring [1,2]. Profitability in the livestock industry can be optimized by considering production efficiency characteristics, particularly with fetal growth and development and postnatal performance [3]. Growth and carcass characteristics of animals differ notably even when genetics and diet are constant. These variations have been attributed to differences in endogenous and exogenous hormones, general health, and immune system status [4]. Most epidemiological studies have shown that inadequate maternal nutrition during gestation can compromise the pregnancy. Inadequate maternal nutrition in humans, rats, and sheep leads to decreased fetal growth and development, predisposing offspring to metabolic, cardiovascular, and endocrine diseases after birth [5]. Fetal growth restriction (FGR) is the failure of a fetus to realize its normal growth potential. FGR is most commonly caused by placental dysfunction and is the second most common cause of perinatal morbidity and mortality [6].

Cardiomyocytes are individual functional cells of cardiac muscles, providing the contractile power of the heart that enables the pumping of blood around the body [7]. Several studies indicate that maternal nutrient restriction increases both left and right ventricular weights in fetal hearts compared with controls [8,9]. FGR, especially in the last three months, can alter the vasoreactivity of coronary arteries and cardiomyocyte maturation, enlargement, and proliferation in the fetus, which may be an indication of later life cardiovascular dysfunction [10]. Therefore, it is important to understand how the cardiac muscle grows in early life and how it responds to cardiovascular disease and injury.

Several studies have examined potential therapeutic supplementation strategies, such as folate, cobalamin, pyridoxine, and melatonin, to enhance placental and offspring development in compromised pregnancies [11]. Melatonin is an indoleamine hormone produced by the pineal gland and secreted in a circadian rhythm. This hormone is well-known as a sleep inducer and regulator of the circadian rhythm as well as different important physiological functions, such as acting as a free radical scavenger, anti-inflammatory, antioxidant, and anticarcinogenic [12]. Melatonin plays a role in the regulation of many physiological systems, including the cardiovascular system, influences myocardial contractility, and blood pressure and melatonin receptors were discovered in the heart and arteries [12]. Some evidence indicates that melatonin is neuroprotective and has a positive impact on the outcomes of compromised pregnancies beginning with the oocyte quality and finishing with parturition [13]. Unfer et al. [14] showed that melatonin administration prior to in vitro fertilization cycles and during pregnancy improved the outcome of the pregnancies. Moreover, maternal treatment with melatonin significantly enhanced placental antioxidant enzyme gene expression [15].

Melatonin concentrations increase in maternal blood during pregnancy, and the presence of melatonin has been demonstrated in amniotic fluid [16]. Maternal melatonin influences the fetal circadian rhythms by providing photoperiod information to the offspring. In addition, maternal melatonin supplementation has been shown to decrease fetal hypoxia, brain injury, and oxidative stress, and improve neurodevelopment in animal models [17]. Most of these animal model results have used a type of mixed-effects model to estimate the effect of variables. Therefore, traditional animal studies with parametric models can have restrictive assumptions while new machine learning techniques are easier to specify. Machine learning has emerged with big data technologies and high-performance computing to create new possibilities for prediction and inference. Learning from data is at the core of machine learning [18]. Machine learning techniques involve a process of learning from training data to perform a task. Data in machine learning consist of a set of examples. Typically, an individual example is described by a set of features or variables. Usually, machine learning tasks are classified into different broad categories depending on the learning type (supervised/unsupervised), learning models (classification, regression, clustering, and dimensionality reduction), or the learning models employed to implement the selected task [19].

This study also leverages the veracity and volume of data available to evaluate whether machine learning models can be utilized to make predictions of hypertrophy and binucleation features of the dataset. To this end, the correlation and importance of each feature in the dataset were evaluated to inform the design of feature representation for machine learning. Furthermore, various model architectures are investigated in terms of their performance in predicting hypertrophy and binucleation from other parameters in the dataset.

The primary objective of this study was to examine the effects of feed restriction and melatonin supplementation on fetal cardiomyocyte cell development parameters in pregnant beef heifers. The secondary objective was to evaluate the fetal cardiomyocyte maturation and enlargement to determine potential fetal programming that may be associated with maternal melatonin supplementation in a bovine model of FGR. In addition, the traditional linear model of comparison between groups and the most common machine learning techniques are compared to find the most accurate method to predict cardiomyocyte development and maturation influenced by feed restriction and melatonin supplementation.

## 2. Materials and Methods

### 2.1. Animal and Experimental Design

Eighty Brangus heifers were artificially inseminated using a single sire at the H. H. Leveck Animal Research Center (Starkville, MS, USA). Animal management and developmental programming results are further presented in [20]. Pregnancy was confirmed by day 35 post-insemination and only singleton bearing heifers were enrolled in the study. All heifers were fed chopped grass hay and a mineral and vitamin supplement to meet 100% NE recommendations for maternal and fetal growth (NRC, 2000) and to meet or exceed mineral and vitamin recommendations during the first half of pregnancy. Twenty-nine pregnant heifers were randomly selected for the study and stratified by maternal body weight recorded on day 140 of gestation. In addition, heifers were trained to acquire feed from a Calan feeding system during this time. On day 160 of gestation, heifers were assigned to one of four treatment groups in a 2 × 2 factorial design: (1) 100% of nutrient requirements (adequately fed; ADQ) with no dietary melatonin (CON); (2) 100% of nutrient requirements (ADQ) with 20 mg of dietary melatonin (MEL); (3) 60% of nutrient requirements (nutrient-restricted; RES) with no dietary melatonin (CON); (4) 60% of nutrient requirements (RES) with 20 mg of dietary melatonin (MEL). This resulted in four treatment groups consisting of ADQ-CON (n = 7), ADQ-MEL (n = 7), RES-CON (n = 7), and RES-MEL (n = 8). Nutrient-restricted heifers received 60% of the same control TMR diet. Heifer body weight was recorded every week and feed was adjusted to achieve the proper average daily gain for pregnancy. Melatonin-supplemented heifers received 20 mg of melatonin similar to previously published methods [21]. Briefly, melatonin was dissolved in ethanol at a concentration of 10 mg of melatonin per mL. Two lb. of grain was top-dressed with 2 mL of ethanol containing 10 mg/mL melatonin or 2 mL of ethanol containing no melatonin (for the control non-melatonin-supplemented group). All heifers received this grain at 0900 h during the treatment period. Following grain consumption (10–15 min), all heifers received their TMR diet at their prescribed nutritional plane. Nutritional plane and treatments were maintained from day 160 to 240 of gestation.

### 2.2. Gestational Day 240 Measurements

On day 240 of gestation, 80 days post-treatment initiation, heifers underwent a Cesarean section. The fetus was weighed and exsanguinated. Fetal heart weight and ventricle thickness were determined for direct comparison. Average left and right ventricular thickness were assessed with digital calipers. A 1 cm by 1 cm heart ventricle section was placed in optimal cutting temperature (OCT) tissue embedding media (Fisher Scientific, Pittsburgh, PA, USA), snap frozen, and stored at −80 °C for later processing.

### 2.3. Heart Immunofluorescence Imaging

Hearts embedded in OCT molds were sectioned using a CRYOSTAR NX50 (Thermo Scientific, Waltham, MA, USA) into seven 10-μm to visualize the cell wall and four 10-μm sections to visualize the nuclei and capillary labeling and mounted on glass slides. For immunofluorescence imaging of capillary and artery, slides were taken out of the freezer 2–3 h before staining, then rinsed three times with TBS 1% for 10 min each. The slides were incubated for 1 h with 10% goat serum in PBS. Slides were treated with the primary antibody, CD31 (ab28364; Abcam, Cambridge, MA, USA), and incubated at room temperature in humidified boxes overnight. Slides were then rinsed three times with TBS 1% for 10 min each and incubated with secondary antibody, Goat Anti-Rabbit IgG H&L (Alexa Fluor 594; ab150080; Abcam, Cambridge, MA, USA) for 1 h. Next, slides were rinsed three times with TBS 1% for 10 min each and rinsed with tap distilled water for 2 min. Lastly, slides were counterstained with 1–2 drops of Fluoroshield mounting medium with DAPI (ab104139; Abcam, Cambridge, MA) for nuclear staining [22,23]. Cell wall was identified in histological sections using Wheat Germ Agglutinin Alexa Fluor 488 (Invitrogen, Walthan, MA, USA) [7]. Images were captured using an EVOS microscope (AMAFD1000; Life Technologies, Carlsbad, CA, USA) with 20× magnification. Approximately 50 stained cardiomyocytes were counted per animal. Images were analyzed using ImageJ (https://imagej.nih.gov/ij/download.html (access on 11 May 2021). Total large blood vessels, nucleus and capillary number per tissue area (number per mm^2^), percent capillary area, cell wall area and nucleus area (%/mm^2^), average capillary size (μm^2^), capillary perimeter per tissue area (mm/mm^2^ or mm^−1^), large blood vessel perimeter (or boundary), and diameter length were recorded. Number, length (μm), size (μm^2^), and perimeter (μm) were recorded for mononucleated and binucleated cardiac muscle cells and mononucleated and binucleated nuclei. Large blood vessels including arterioles and veins were differentiated from capillaries according to the thickness of their walls. Morphologically, large blood vessels have thicker walls compared to the extremely thin walls in capillaries. Capillaries are very small and require higher magnification than arterioles and veins.

### 2.4. Training Machine Learning Models

Prior to training machine learning models, the dataset was pre-processed by computing the standard deviation, median, and average of the multiple measurements of numerical features for each cow to generate unified data points. Additionally, the dependent features that were formed from two or more features were removed. The numerical target variable of data points was converted into two classes by thresholding via taking the average of the target variable, hypertrophy, and binucleation and assigning the data points with 0 (less than average) or 1 (greater than average). Then, the data were split into 80% of data for training the model and 20% for testing the model performance.

### 2.5. Statistical Analysis

The effects of diet on dependent variables were tested with the MIXED procedure of SAS (SAS software version 9.4, SAS Institute, Cary, NC, USA). Dependent variables measured over time were analyzed using repeated-measures ANOVA of the MIXED procedure of SAS, and means were separated using the PDIFF option of the LSMEANS statement. The model statement included: melatonin treatment, nutritional plane, and their interaction, day of gestation, and fetal sex. Least square means and SE were reported. Statistical significance was declared at *p* ≤ 0.05.

## 3. Results

Maternal body weight results are further explained in Contreras-Correa et al. [20]. Briefly, maternal body weight, fetal body weight, and fetal thoracic girth were decreased (*p* < 0.05) in RES- versus ADQ-fed heifers at day 240 of gestation (Table 1). Melatonin supplementation did not mitigate maternal and fetal body weights. Fetal heart weight was not different between treatment groups; however, fetal heart right ventricle thickness was decreased (*p* = 0.001) by maternal nutrient restriction (Table 1).

Fetal large blood vessel characteristics are illustrated in Figure 1 micrographs. Large blood vessel numbers per tissue area were not different between treatments and averaged 2.03 ± 0.07 (Figure 2A). Large blood vessel size tended to increase in RES vs. ADQ (Figure 2B), but their diameter length was not different between treatments and averaged 23.29 ± 1.12 μm (Figure 2C). Large blood vessel perimeter was increased (*p* = 0.05) in fetuses from RES compared to CON (Figure 2D).

Significant (*p* < 0.05) nutrition by treatment interactions were observed for capillary characteristics in the fetal heart (Table 2). Capillary number per tissue area was increased in RES-MEL versus ADQ-MEL fetuses. Percent capillary area, capillary size, and total capillary perimeter were all decreased in RES-CON, RES-MEL, and ADQ-MEL compared to ADQ-CON fetuses.

Figure 3A shows the percent nucleus area and percent cell wall area in 8-month-old fetuses. The percent nucleus area tended to increase (*p* = 0.08) in RES (17.12 ± 0.25) compared with ADQ (16.50 ± 0.26), while treatment decreased (*p* = 0.05) percent nucleus area in MEL (16.46 ± 0.24) compared with CON (17.16 ± 0.28). Percent cell wall area as demonstrated in Figure 3B was not different between treatments and averaged 44.66 ± 0.91. 

In general, fetal cardiomyocyte characteristics were severely impacted by maternal nutritional plane (Table 3). A major effect of treatment was observed for the nucleus number per tissue area, which decreased (*p* = 0.0004) in MEL (1638 ± 26.45) versus CON (1773 ± 27.12) fetuses (data not shown). Binucleated cell number per tissue area exhibited a nutrition by treatment interaction (*p* = 0.010), which was decreased in ADQ-MEL and RES-CON compared with ADQ-CON fetuses. Percentage of cardiomyocyte binucleation was significantly decreased (*p* = 0.001) in RES (5.12 ± 0.40) fetuses compared with ADQ (7.00 ± 0.40). The binucleated nucleus length was decreased in RES (4.85 ± 0.14 μm) compared with ADQ (5.76 ± 0.14 μm) groups. Mononucleated nucleus length exhibited a nutrition by treatment interaction (*p* = 0.0005), which was decreased in RES-MEL compared with ADQ-MEL, while ADQ-CON and RES-CON were intermediate.

Binucleated cardiac muscle cell length tended (*p* = 0.06) to be decreased in RES (15.07 ± 0.53 μm) versus ADQ (16.47 ± 0.53 μm) fetuses (Table 3). Mononucleated cell length was decreased (*p* < 0.001) in RES versus ADQ fetuses. Furthermore, mononucleated cell length, binucleated nucleus size, mononucleated nucleus size, binucleated nucleus perimeter, and mononucleated nucleus perimeter were all decreased (*p* < 0.001) in RES versus ADQ fetuses (Table 3). A nutrition by treatment interaction (*p* < 0.01) was observed for binucleated and mononucleated cell size, which were increased in fetuses from ADQ-MEL versus all other groups. Cell perimeter in mononucleated cardiac muscle cells was decreased (*p* < 0.0001) in RES versus ADQ (Table 3).

The model performance of all the experimented ML models is shown in Figure 4. From the plots, it appears that the Random Forest and Decision Tree models’ performance is better than the Bagging Classifier, but the mentioned models are overfitting on this small dataset. Since Bagging Classifiers can reduce variance and help to avoid overfitting, it was suitable for our small dataset in order to generate accurate predictions of target variables. 

In this Bagging Classifier model, we considered the Decision Tree machine learning model as a base estimator for predicting both target variables. The “n_estimators” and “random_state” hyperparameters of the model are experimentally adjusted to further enhance the performance of the model. For the binucleation target variable, the Bagging model with 5 Decision Tree estimators produced the best results without overfitting. Similarly, for the hypertrophy target variable, the best Bagging model performance is with 3 Decision Tree estimators. The model performance for predicting both target variables is shown in Table 4.

For each target variable, the importance of each feature for model predictions was calculated. In the prediction of binucleation, the left ventricle thickness feature had the highest Gin importance weight of 0.20 followed by fetal BW with 0.11 weight, whereas, in the case of hypertrophy, heart weight (g) had the highest importance weight of 0.68. The feature importance weight plots for both the target variables are shown in Figure 5.

Furthermore, there was a possibility of increasing the Bagging Classifier model performance if larger data were available and if the data had temporal features. Because of the data size restriction, high-level machine learning models were not experimented in order to obtain accurate predictions.

## 4. Discussion

In the present study, the nutritional plane was shown to have a significant effect on maternal body weight. This result is similar to our previous study, in which maternal body weight and body condition were decreased in an ovine model of intrauterine growth restriction [8]. In addition, fetal body weight and fetal thoracic girth were decreased in RES versus ADQ fetuses. In preliminary studies, melatonin has been associated with pain regulation, sedation effect, and lower anxiety, and it acts as an analgesic in the preoperative period [24]. Thus, melatonin may be related to decreased locomotor activity; however, our previous work did not examine these behavioral changes [8,23]. Melatonin dosages in these experiments were much lower than doses predicted to alter behavior. For example, the analgesic property of melatonin treatment was observed at 20–200 mg/kg of body weight [25], while in the present study heifers received 36 μg/kg of body weight. Similar studies [26] have observed that metabolic BW in control dams increased, whereas metabolic BW in restricted-fed dams decreased. Maternal nutritional intake during pregnancy is imperative for normal fetal growth and has lifelong impacts on the health and performance efficiency of the offspring [27,28]. Adequate nutritional intake during pregnancy is critical for fetal development and physiological consequences on offspring performance [29]. Maternal bodyweight reduction in our study coincided with a reduction in fetal body weight of RES compared with ADQ fetuses. Furthermore, if the intrauterine environment suffers from nutrient restriction, but the postnatal extrauterine environment is nutritionally abundant, then the offspring’s growth parameters may be inappropriately programmed and negatively impact performance in postnatal life [30,31,32,33].

Two basic mechanisms allow the coronary vasculature to raise the myocardial oxygenation level as O2 demand increases: (1) augmentation of blood flow via dilation of the existing resistance vessels and (2) an increase in the number of blood vessels, which is the result of stimulated angiogenesis [34]. Angiogenesis reveals a broad potential to alter the outcome of coronary and peripheral artery diseases such as aneurysms, atherosclerosis, and arteriovenous malformations [35]. Melatonin is involved in regulating angiogenesis. Observations demonstrated that melatonin serves dual roles in the inhibition of angiogenesis, as an antioxidant and a free radical scavenging agent. Fetal growth restriction during late gestation in sheep altered fetal coronary artery vasoconstriction to angiotensin [36]; therefore, maternal undernutrition can alter cardiomyocyte maturation, enlargement, and proliferation, which may cause cardiovascular dysfunction of offspring in later life. Large blood vessel numbers and length were not different between groups; however, large blood vessel size tended to increase in RES groups, and large blood vessel perimeter increased in fetuses from RES dams. These results are similar to previous experiments on maternal nutrition restriction that reported that the lumen diameters of carotid and femoral arteries were not significantly different among groups. Arterial wall thickness was also not different among groups [37]. However, nutrition restriction reduced fetal growth ventricular size and increased relative to whole body weight [38]. Melatonin treatment did not change large blood vessel characteristics in our study, while previous reviews reported that an amplitude of melatonin secretion altered cardiovascular function and has been associated with an improvement of oxidative status steroid and prostaglandin metabolism in rats and sheep [39,40,41,42]. Melatonin restricts the viability and angiogenesis of vascular endothelial cells by suppressing vascular endothelial growth factor, reactive oxygen species, and hypoxia-inducible factor [43], which should be investigated in future studies.

The function of capillaries is the delivery of oxygen and removal of metabolites to or from tissues. In this context, increased capillary surface area results in greater potential for diffusion of oxygen and metabolite clearance, with improving endurance and aerobic capacity [44]. Some experiments reported a correlation between capillary density and average mitochondrial volume density among different skeletal muscles that considered oxidative fibers [44]. In the present study, maternal melatonin supplementation decreased the percent capillary area, capillary size, and capillary perimeter, consistent with our previous results [8]. However, nutritional plane increased the capillary number area in RES, while the capillary size and capillary perimeter were decreased in RES groups. These opposite responses could be related to chronic increases in artery blood flow in ADQ groups, leading to alterations in the percent capillary area in offspring. These results corroborate previous reports that maternal protein restriction in pregnant rats results in decreased numbers of cardiomyocytes [45]. In contrast, previous experiments on the role of capillary disorder in coronary ischemic congestive heart failure (CHF) determined that capillary density increased in remote myocardium. In this study, capillaries in CHF hearts increased from 43 to 83% [46]. However, the role of capillaries in the development of CHF is not fully understood at present. Epidemiological studies have shown that subjects exposed to disruption in early development have an increased incidence and prevalence of cardiovascular disease in the future [47]. It is likely that a limiting intrauterine environment may permanently reduce the numbers of functional units in vital tissues and organs, resulting in altered function and influencing postnatal organ function later in life. Compromised pregnancies in numerous animal models decreased vascularity due to underfeeding or overfeeding adolescents, underfeeding adults, or multiple pregnancies [48]. It is assumed that a decrease in the functional units of cardiomyocytes in fetal hearts will limit postnatal cardiac growth capacity. Therefore, when cardiomyocytes reach their limits of hypertrophy, further heart enlargement will occur through increased deposition of the extracellular matrix and lead to cardiac fibrosis. Similarly, our experiment showed that decreasing capillary numbers was associated with increasing capillary size and perimeter.

Interestingly, in our study, nucleus number increased in fetuses from melatonin-treated heifers. Previous studies have shown that melatonin free radical scavenging activities are accomplished without interaction with a receptor and modified cell physiology via the membrane and possibly nuclear receptors [49]. We propose that compromised pregnancies will increase melatonin receptors in the nuclei of cells to raise immunoreactivity and inflammatory responses of melatonin. These responses may be related to the nucleus numbers, which increased in fetuses from melatonin dams. This hypothesis is supported by our experimental data, according to which the nucleus number was not affected by nutritional plane, while in a separate study, the number of cardiomyocyte nuclei reduced in the maternal protein restriction group [50].

Studies have reported that abnormalities in cardiac growth have their roots in the early stages of life [51]. Cardiomyocyte nuclearity is imperative in studies of myocardial development and disease. In many species, binucleation is a sign of cardiomyocyte maturity (except in humans or other primates). The majority of cardiomyocytes in fetuses were mononucleated. Binucleated cardiomyocytes had larger size and length than mononucleated cardiomyocytes. In the ADQ groups, binucleated percentages were greater than in RES groups, whereas the binucleated cardiomyocyte number and binucleated percentage exhibited a melatonin treatment by nutritional plan interaction, and they were increased in fetuses from RES-MEL versus RES-CON dams. As cardiomyocyte proliferation is rare after birth, it is probable that reduction in cardiomyocyte numbers due to maternal nutrient restriction may lead to compromised cardiac function in later life. The number of binucleated cardiomyocytes was low and averaged 6.05 + 1.24% between the groups. Similar to our result, Corstius et al. [50] reported an approximately 3% binucleated percentage, which was not different between offspring from the low-protein diet or control rats. These data contrast with previous reports in which the binucleated percentage in weanling mice, fetal sheep, and 9-week-old lambs were 80 + 1.4%, 87.1 + 1.6%, and 97.6 + 0.9%, respectively [7].

## 5. Conclusions

Maternal nutrition restriction decreased body weight in pregnant heifers and their fetuses. This was associated with decreasing thoracic girth but was not accompanied by a significant decrease in heart weight, which may be a mark of heart hypertrophy. Melatonin treatment did not save fetal weight in nutrient-restricted groups. To our knowledge, this is the first study that has reported the effect of FGR and melatonin treatment on the cardiomyocyte parameters in the heart. This study has provided evidence that compromised pregnancy leads to a reduction in the number of cardiomyocytes, while melatonin treatment can mitigate some of these disturbances. For the binucleation and hypertrophy target variables, the Bagging model with 5 Decision Tree estimators and 3 Decision Tree estimators produced the best results without overfitting. In the prediction of binucleation, the left ventricle thickness feature had the highest Gin importance weight followed by fetal BW. In the case of hypertrophy, heart weight (g) had the highest importance.

## Figures and Tables

**Figure 1 animals-12-01818-f001:**

Representative immunofluorescence images of 8-month old fetal cardiomyocytes. Cryosectioned cardiomyocytes were stained for (**A**) cell wall using WGA (Alexa Fluor 488) and nuclei (blue, DAPI) with yellow circle showing binucliated cell, (**B**) capillaries, and (**C**) large blood vessels with CD31 (red, Alexa Fluor 594)). The white scale bar represents 200 μm for figures **A** and **B** and 400 μm for figure **C**.

**Figure 2 animals-12-01818-f002:**
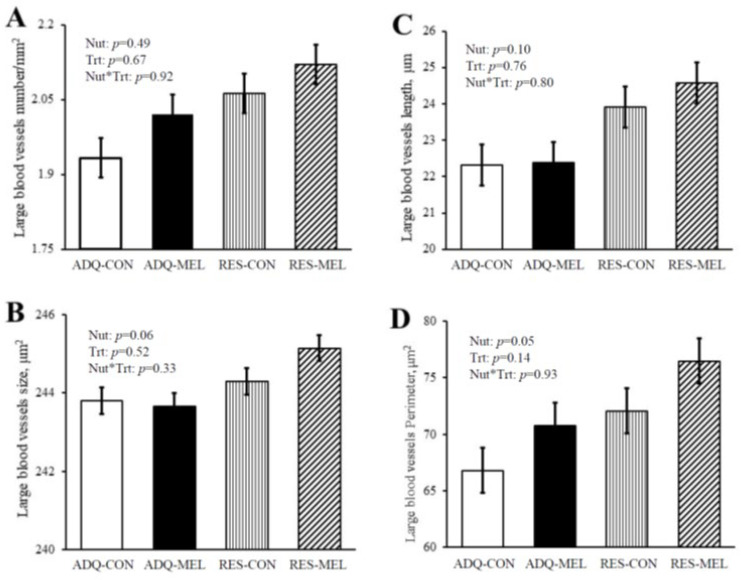
Large blood vessel characteristic of 8-month-old fetuses. (**A**) Large blood vessel numbers per tissue area, (**B**) large blood vessel size, (**C**) large blood vessel diameter length, and (**D**) large blood vessel perimeter. Values are means ± SE.

**Figure 3 animals-12-01818-f003:**
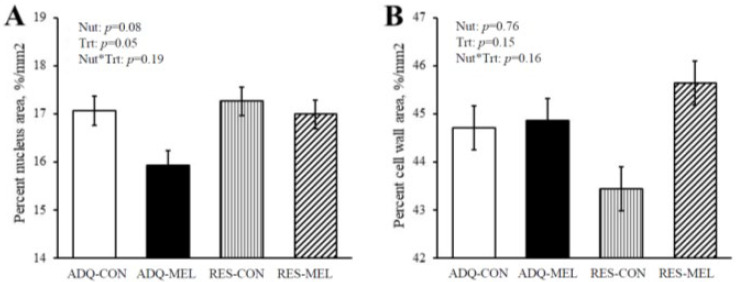
Percent area of nucleus (**A**) and Percent cell wall area (**B**) of 8-month-old fetuses. Values are means ± SE.

**Figure 4 animals-12-01818-f004:**
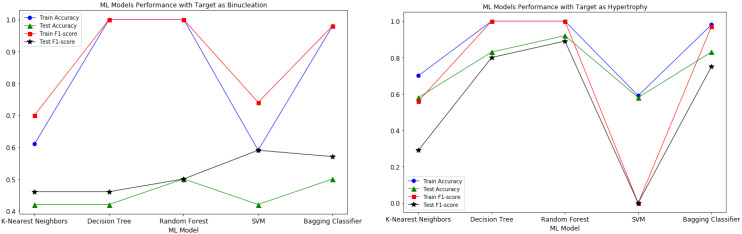
Model Performance metrics comparison plots for both the target variables with K-Nearest Neighbors, Decision Tree, Random Forest, SVM and Bagging Classifier Models.

**Figure 5 animals-12-01818-f005:**
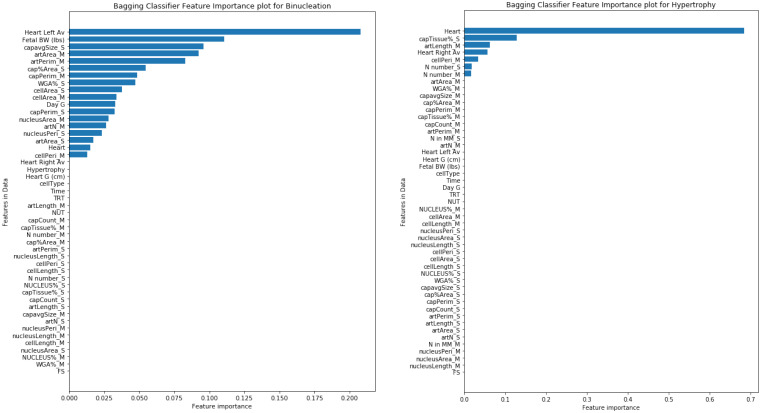
Importance of each feature during the prediction of target variables Binucleation & Hypertrophy with Bagging Classifier.

**Table 1 animals-12-01818-t001:** Maternal body weight and fetal growth parameters from day 240 of gestation. Data adapted from Contreras-Correa et al. [20].

Dependent Variable	ADQ-CON	ADQ-MEL	RES-CON	RES-MEL	SE	*p*-Value		
						Nut	trt	Nut*trt
Maternal BW (kg)	575.7	580.2	531.8	536.2	18.8	0.01	0.87	0.99
Fetal BW (kg)	26.7	26.9	23.3	24.6	1.0	0.01	0.23	0.62
Thoracic girth (cm)	64.2	64.2	60.9	62.7	0.9	0.01	0.31	0.35
Heart weight (g)	186.2	187.2	173.7	174.2	9.6	0.17	0.94	0.98
Left ventricle thickness (mm)	10.4	11.1	11.1	10.7	0.8	0.84	0.83	0.48
Right ventricle thickness (mm)	7.2	7.8	5.8	6.4	0.4	0.001	0.12	0.96
Heart thickness/heart weight	0.09	0.10	0.10	0.10	0.05	0.69	0.27	0.82

**Table 2 animals-12-01818-t002:** Histological analysis for capillary characteristics of 8-month-old fetuses. Different letter superscripts (a–c) represent significant differences (*p* < 0.05) between the nutritional plane and treatment interaction.

Dependent Variable	ADQ-CON	ADQ-MEL	RES-CON	RES-MEL	SE	*p*-Value		
						Nut	Trt	Nut*trt
Capillary number per mm^2^	1142 ^ab^	1106 ^b^	1139 ^ab^	1172 ^a^	15.47	0.03	0.93	0.01
Percent capillary area, %/mm^2^	7.01 ^a^	6.08 ^c^	6.49 ^b^	6.48 ^bc^	0.15	0.70	0.002	0.0023
Capillary size, μm^2^	20.77 ^a^	19.02 ^b^	18.80 ^b^	19.17 ^b^	0.36	0.01	0.05	0.002
Total capillary perimeter (mm^−1^)	90.07 ^a^	79.36 ^c^	85.04 ^b^	83.69 ^b^	1.82	0.84	0.0007	0.0078

**Table 3 animals-12-01818-t003:** Cardiomyocyte characteristics of 8-month-old fetuses. Different letter superscripts (a–d) represent significant differences (*p* < 0.05) between the nutritional plane and treatment interaction.

Dependent Variable	ADQ-CON	ADQ-MEL	RES-CON	RES-MEL	SE	*p*-Value		
						Nut	Trt	Nut*trt
Binucleated cells number per mm^2^	388 ^a^	284 ^b^	281 ^b^	322 ^ab^	27.28	0.21	0.25	0.010
Cardiomyocyte binucleation %	7.66	6.33	4.78	5.46	0.59	0.001	0.58	0.054
Binucleated nucleus length (μm)	5.76	5.76	4.86	4.85	0.20	<0.0001	0.98	0.98
Mononucleated nucleus length (μm)	5.77 ^b^	6.01 ^a^	5.62 ^b^	5.28 ^c^	0.087	<0.0001	0.56	0.0005
Binucleated cell length (μm)	15.96	16.98	15.98	14.16	0.81	0.061	0.58	0.07
Mononucleated cell length (μm)	11.72	11.99	10.53	10.22	0.20	<0.0001	0.93	0.11
Binucleated nucleus size (μm^2^)	18.91	20.33	15.59	14.89	0.83	<0.0001	0.66	0.14
Mononucleated nucleus size (μm^2^)	19.75	19.40	16.46	14.67	0.45	<0.0001	0.015	0.06
Binucleated nucleus perimeter (μm)	16.54	17.05	15.27	14.75	0.37	<0.0001	0.99	0.11
Mononucleated nucleus perimeter (μm)	17.14	16.89	15.80	14.73	0.20	<0.0001	0.0010	0.019
Binucleated cell size (μm^2^)	91.44 ^b^	121.16 ^a^	80.79 ^b^	77.48 ^b^	6.73	<0.0001	0.05	0.005
Mononucleated cell size (μm^2^)	60.33 ^b^	66.52 ^a^	51.12 ^cd^	47.53 ^d^	1.71	<0.0001	0.43	0.0008
Binucleated cell perimeter (μm)	41.62 ^b^	46.28 ^a^	38.17 ^bc^	36.62 ^c^	1.61	<0.0001	0.34	0.02
Mononucleated cell perimeter (μm)	32.64 ^a^	33.76 ^a^	29.84 ^b^	28.87 ^b^	0.50	<0.0001	0.88	0.015

**Table 4 animals-12-01818-t004:** Bagging Classifier Performance Results (* Binucleated cells number per mm^2^ of tissue).

Target Variable	Accuracy		F1-Score		ROC AUC	
	Training	Test	Training	Test	Training	Test
Hypertrophy	0.935	0.833	0.919	0.800	0.929	0.829
N in mm * (Average)	0.978	0.750	0.981	0.571	0.981	0.700

## Data Availability

The data presented in this study is available on reasonable request from the corresponding author.

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
