# Peer review of "Effects of Maternal Nutrient Restriction and Melatonin Supplementation on Cardiomyocyte Cell Development Parameters Using Machine Learning Techniques"

_animals, 2022, doi:10.3390/ani12141818_

Round 1

Reviewer 1 Report

The manuscript titled Effects of maternal nutrient restriction and melatonin supplementation on cardiomyocyte development parameters using machine learning techniques was difficult to review. While the “machine learning” is in the title and an objective of the study this part is lost throughout the results, and frankly seems as another statistical analysis. It is not clear why the variables were chosen for the machine learning, and really the results are expected—for hypertrophy heart weight had the most influence. I don’t have any suggestions for making this better, but as the reader, this part is very unclear. If this is the objective of the study (it is), these results should be presented first in the results and not appear as an after thought. Overall it is unclear what the paper is about—is it about melatonin supplement in a restricted nutrient environment or is about machine learning. It would help the readability of the manuscript if the authors were more clear on the objective and wrote the paper as so. The discussion should spend some time discussing Figure 4 plots. To the reader, these are very different yet this is not addressed in the manuscript.

Other considerations

Line 187-190. Result and should be in the results section not in the methods.

The first paragraph of the discussion needs to be deleted.

Author Response

Comments and Suggestions for Authors

The manuscript titled Effects of maternal nutrient restriction and melatonin supplementation on cardiomyocyte development parameters using machine learning techniques was difficult to review. While the “machine learning” is in the title and an objective of the study this part is lost throughout the results, and frankly seems as another statistical analysis. It is not clear why the variables were chosen for the machine learning, and really the results are expected—for hypertrophy heart weight had the most influence. I don’t have any suggestions for making this better, but as the reader, this part is very unclear. If this is the objective of the study (it is), these results should be presented first in the results and not appear as an after thought. Overall it is unclear what the paper is about—is it about melatonin supplement in a restricted nutrient environment or is about machine learning. It would help the readability of the manuscript if the authors were more clear on the objective and wrote the paper as so. The discussion should spend some time discussing Figure 4 plots. To the reader, these are very different yet this is not addressed in the manuscript.

Other considerations

Line 187-190. Result and should be in the results section not in the methods.

The first paragraph of the discussion needs to be deleted.

Reply to reviewer 1: Thanks for your valuable time and reviewing this manuscript, all of your comments are appreciated and we tried to subside unlikeable deficiencies in the paper.

Responding some of your comments, objective of study was modified at the end of introduction. For make it clearer the title of paper was modified as well. However, we mentioned and showed partially use of Machine learning for hypotrophy objection, it was minor objective. Because we tended to have a start for comparing different ML methods to see which one is more fit to such data and for future target work more on it. Therefore, the parameter chosen for this case were more relevant to binucleation and hypertrophy based on litterateur review. The other comments were added to the text.

Reviewer 2 Report

The manuscript entitled "Effects of maternal nutrient restriction and melatonin supplementation on cardiomyocyte cell development parameters using machine learning techniques" was reviewed. The manuscript contains a number of interesting data. However, it is not written well. A number of careless mistakes make the manuscript unacceptable. For example, please see page 11, line 327, "in in" repeatedly written.

Also, Page 11 Line 330, "is not is not".

Its difficult to follow the manuscript because of so many abbreviations.  It is better to avoid the short forms of words/phrases.

Page 2 Line 49-52 and also Line 85-89, please revise for easy understanding of readers

Page 11 Line 360- intake "levels are a critical"..... Please check English

Page 11 Line 359-"fetal nutritional need to fully understood". Please check English.

Page 11 Line 356 Rewrite for easy understanding

Page 11 Line 355 Similar studies by [29] observed increased meta.... Reference style was not followed properly

Page 11 Line 350: Lemley et al. (2012) nor in McCarty et al. (2018) : Follow the style for citation.

In materials and methods, Reference style was not followed.

Page 11 line 349 What is IUGR?

Page 4 Line 141-143 should be deleted as these are results not methodology

What is "machine learning technique" ? Its not understood clearly in this manuscript. It would be better to consider that the readers would not understand these easily.

A sentence should not be started with a number. Please see the first sentence of result section

Fig 4 is not visible properly.

 Table 4 should be deleted, and its better to explain its in the text.

Page 6 Line 202, table 1 should be Table 1

Tables and Figures must be self explanatory.  Its hard to follow the data due to use of so many abbreviations.

figure 2A should be Figure 2A, figure 2D- Figure 2D: see page number 6 Line Line 214, 215, 217. Please follow journal style. From Line 212 to 218, English should be edited. "Blood vessels number" should be "blood vessel numbers"

Conclusion is too long.

Author Response

The manuscript entitled "Effects of maternal nutrient restriction and melatonin supplementation on cardiomyocyte cell development parameters using machine learning techniques" was reviewed. The manuscript contains a number of interesting data. However, it is not written well. A number of careless mistakes make the manuscript unacceptable. For example, please see page 11, line 327, "in in" repeatedly written.

Reply to reviewer 2: Thanks for your precise review, we are so honored to have your comments. We revised the manuscript after considering your comments as much as possible and thanks again for helping us to improve it.

Also, Page 11 Line 330, "is not is not".

Reply to reviewer 2: Thanks for your comment, we revised it.

Its difficult to follow the manuscript because of so many abbreviations.  It is better to avoid the short forms of words/phrases.

Page 2 Line 49-52 and also Line 85-89, please revise for easy understanding of readers

Reply to reviewer 2: Thanks for your comment, we revised it.

Page 11 Line 360- intake "levels are a critical"..... Please check English

Reply to reviewer 2: Thanks for your comment, we revised it.

Page 11 Line 359-"fetal nutritional need to fully understood". Please check English.

Reply to reviewer 2: Thanks for your comment, we revised it.

Page 11 Line 356 Rewrite for easy understanding

Reply to reviewer 2: Thanks for your comment, we revised it.

Page 11 Line 355 Similar studies by [29] observed increased meta.... Reference style was not followed properly

Reply to reviewer 2: Thanks for your comment, we revised it.

Page 11 Line 350: Lemley et al. (2012) nor in McCarty et al. (2018) : Follow the style for citation.

Reply to reviewer 2: Thanks for your comment, we revised it.

In materials and methods, Reference style was not followed.

Reply to reviewer 2: Thanks for your comment, we revised it.

Page 11 line 349 What is IUGR? Reply to reviewer 2: Thanks for your comment, we removed it and used the extended term instead.  

Page 4 Line 141-143 should be deleted as these are results not methodology

Reply to reviewer 2: Thanks for your comment, we revised it.

What is "machine learning technique" ? Its not understood clearly in this manuscript. It would be better to consider that the readers would not understand these easily.

Reply to reviewer 2: They were different ML techniques using in this paper to have a comparison between them and figure out which one is best for animal studies. Therefore different machine learning model architectures like K-Nearest Neighbors, Decision Tree, Random Forest, Support Vector Machine (SVM) were used. It was mentioned at abstract first and then in objection and discussion again.

A sentence should not be started with a number. Please see the first sentence of result section

Reply to reviewer 2: Thanks for your comment, we revised it.

Fig 4 is not visible properly.

 Table 4 should be deleted, and its better to explain its in the text.

Reply to reviewer 2: Thanks for your comment, it was explained above the table however it might be more accurate for readers to now exact numbers at the table.

Page 6 Line 202, table 1 should be Table 1

Reply to reviewer 2: Thanks for your comment, we revised it.

Tables and Figures must be self explanatory.  Its hard to follow the data due to use of so many abbreviations.

figure 2A should be Figure 2A, figure 2D- Figure 2D: see page number 6 Line Line 214, 215, 217. Please follow journal style. From Line 212 to 218, English should be edited. "Blood vessels number" should be "blood vessel numbers"

Reply to reviewer 2: Thanks for your comment, we revised it.

Conclusion is too long.

Round 2

Reviewer 1 Report

The manuscript titled Effects of maternal nutrient restriction and melatonin supplementation on cardiomyocyte cell development parameters using machine learning techniques by Mazinani has been revised and is somewhat improved. The way the results are presented is completely confusing—it is cumbersome to go from the text to the tables—a casual reader would be lost. I think the manuscript would be greatly improved if the results stated some global observations at the beginning—following the obvious that maternal nutrition decreased maternal and fetal body weight it could be stated that while heart weight was not affected by nutrition plane right ventricle thickness (Table 1), capillary size (Table 2), and measures of cardiomyocytes were reduced by maternal feed restriction—these were largely unmitigated by melatonin treatment (Table 3). Following the summary results, interactions could be presented in a way that a casual reader can follow and make sense of these results. I am strongly suggesting the results be rewritten.

Other considerations.

Line 27. There is not a comparison between RES and CON since RES is the nutritional plane and CON is the melatonin treatment. As per Figure 2 –differences are by nutritional plane so ADQ vs RES. Revise.

Line 36. It is not clear this is a compromised pregnancy. I suggest “restricted maternal nutrition”.

Line 84 – 86. Sentence is confusing. Revise for clarity.

Line 134. Please add how this differed among the nutritional treatments. It seems that the ADQ group was adjusted for proper ADG, and the RES group was adjusted relative to that. Please clarify.

Line 155. Was each of three rinses 10 minutes, or was the three rinses within 10 minutes. Clarify.

Line 156. Add antibody details. Dilution and volume for both primary and secondary antibody.

Line 170. I am unsure what “perimeter and length of diameter” means. Please revise for clarity.

Line 176. Suggest “require” rather than “need”  “[require] higher magnification [than] arterioles and veins.”

Line 215. This is a nutrition tendency (Figure 2B) so would be “RES vs ADQ”. And also, in the paragraph without line numbers following Figure 2 legend (second line).

Line 232. Suggest adding this data to Table 3.

Line 234. Data in Table 3—please indicate.

Line 235-236 also 252-253. Example of very confusing data presentation. The interaction is stated followed by effect of nutrition.  Results must be rewritten for clarity.

Line 388. Do you mean “nutritional plane” not “plan”.

Line 391-392. Vague—please revise for clarity.

Line 416. Suggest “mitigate” rather than “subside”

Author Response

The manuscript titled Effects of maternal nutrient restriction and melatonin supplementation on cardiomyocyte cell development parameters using machine learning techniques by Mazinani has been revised and is somewhat improved. The way the results are presented is completely confusing—it is cumbersome to go from the text to the tables—a casual reader would be lost. I think the manuscript would be greatly improved if the results stated some global observations at the beginning—following the obvious that maternal nutrition decreased maternal and fetal body weight it could be stated that while heart weight was not affected by nutrition plane right ventricle thickness (Table 1), capillary size (Table 2), and measures of cardiomyocytes were reduced by maternal feed restriction—these were largely unmitigated by melatonin treatment (Table 3). Following the summary results, interactions could be presented in a way that a casual reader can follow and make sense of these results. I am strongly suggesting the results be rewritten.

Author: Significant changes were made to results presentation as recommended by the reviewer. The results were summarized as suggested and re-written for clarity. 

Other considerations.

Line 27. There is not a comparison between RES and CON since RES is the nutritional plane and CON is the melatonin treatment. As per Figure 2 –differences are by nutritional plane so ADQ vs RES. Revise.

Author: Corrected in response to reviewer’s comment.

Line 36. It is not clear this is a compromised pregnancy. I suggest “restricted maternal nutrition”.

Author: Corrected in response to reviewer’s comment.

Line 84 – 86. Sentence is confusing. Revise for clarity.

Author: Sentences were revised for clarity.

Line 134. Please add how this differed among the nutritional treatments. It seems that the ADQ group was adjusted for proper ADG, and the RES group was adjusted relative to that. Please clarify.

Author: This is clarified under lines 141-142, “Heifer body weight was recorded every week and feed offered was adjusted to achieve the proper average daily gain for pregnancy”

Line 155. Was each of three rinses 10 minutes, or was the three rinses within 10 minutes. Clarify.

Author: Corrected in response to reviewer’s comment.

Line 156. Add antibody details. Dilution and volume for both primary and secondary antibody.

Author: Changes were made in response to reviewer’s comment; however, specific dilutions are omitted as these need to be tested with each batch of antibody. 

Line 170. I am unsure what “perimeter and length of diameter” means. Please revise for clarity.

Author: Corrected in response to reviewer’s comment

Line 176. Suggest “require” rather than “need”  “[require] higher magnification [than] arterioles and veins.”

Author: Corrected in response to reviewer’s comment

Line 215. This is a nutrition tendency (Figure 2B) so would be “RES vs ADQ”. And also, in the paragraph without line numbers following Figure 2 legend (second line).

Author: Corrected in response to reviewer’s comment

Line 232. Suggest adding this data to Table 3.

Author: Results text for table 3 was rewritten for clarity 

Line 234. Data in Table 3—please indicate.

Author: Results text for table 3 was rewritten for clarity 

Line 235-236 also 252-253. Example of very confusing data presentation. The interaction is stated followed by effect of nutrition.  Results must be rewritten for clarity.

Line 388. Do you mean “nutritional plane” not “plan”.

Author: Corrected in response to reviewer’s comment

Line 391-392. Vague—please revise for clarity.

Author: Corrected in response to reviewer’s comment

Line 416. Suggest “mitigate” rather than “subside”

Author: Corrected in response to reviewer’s comment